# Neural Trajectory Analysis of Recurrent Neural Network In Handwriting Synthesis

**Kristof B. Charbonneau** [*]
École de technologie supérieure
Montreal, Québec, Canada
`kristof.boucher-charbonneau.1`
`@ens.etsmtl.ca`

**Osamu Shouno**
Honda Research Institute Japan, Co., Ltd.
Wako, Saitama, Japan.
`shouno@jp.honda-ri.com`

## Abstract

Recurrent neural networks (RNNs) are capable of learning to generate highly realistic, online handwritings in a wide variety of styles from a given text sequence. Furthermore, the networks can generate handwritings in the style of a particular writer when the network states are primed with a real sequence of pen movements from the writer. However, how populations of neurons in the RNN collectively achieve such performance still remains poorly understood. To tackle this problem, we investigated learned representations in RNNs by extracting low-dimensional, neural trajectories that summarize the activity of a population of neurons in the network during individual syntheses of handwritings. The neural trajectories show that different writing styles are encoded in different subspaces inside an internal space of the network. Within each subspace, different characters of the same style are represented as different state dynamics. These results demonstrate the effectiveness of analyzing the neural trajectory for intuitive understanding of how the RNNs work.

## 1 Introduction

Recurrent neural networks (RNNs) have successfully demonstrated their incredible performances in a wide variety of tasks related to sequential data, such as language modeling, text generation, online handwriting synthesis, speech generation, image captioning, and video prediction. Long Short-Term Memory (LSTM) Hochreiter & Schmidhuber (1997) and derivatives are critical neuronal models for leaning these tasks because of their ability to link events separated by long intervals. However, knowledge of these cellular mechanisms is not sufficient for interpreting learned internal processes in the networks that underly their incredible performance.

There are very few studies investigating internal processes in RNNs. Karpathy et al (2015) were one of the first to investigate the learned representations in RNNs, more specifically from character-level language models, and revealed that cells are encoding specific character patterns. Carter et al (2016) showed examples of neurons in RNNs for online handwriting generation that are sensitive to specific pen movements. Although these studies provide interesting information about single-cell-level representations, an understanding of how populations of neurons in the networks collectively embody incredible capabilities still remains to be explored.

Here we approach this problem by using neural trajectory analyses developed in neurophysiology for understanding activity of the large population of recorded neurons in the brain Yu et. al.(2009). Concretely, we extracted low-dimensional neural trajectories of populations of neurons in the RNNs for online handwriting synthesis (Graves, 2013). Our analysis of neural trajectories reveals that different handwriting styles are represented as different subspaces in an internal neural space and characters of a given style are embedded as state dynamics in the subspace of the style.

---

[*]This work was done during his internship at Honda Research Institute Japan, Co., Ltd.

## 2 EXPERIMENTAL SETUP

### 2.1 RECURRENT NEURAL NETWORK MODEL

The RNN model for the online handwriting synthesis Graves (2013) is trained to predict the next pen state from past states conditioned on a target character sequence from the IAM online handwriting database Liwicki and Bunke (2005). The architecture of the model is composed of three stacked recurrent layers all of which receive real-valued pen-state information, and their outputs are given to the mixture density network which predicts the next pen state. The three recurrent layers also receive inputs from the character sequence mediated by an attention module. Each recurrent layer consists of 400 LSTM units. To sample handwriting in the style of a particular writer, we used the primed sampling method empirically found in Graves (2013) in which the network is first clamped to a real time-series data of pen states and its target character sequence and then handwriting of a given target text is sampled.

### 2.2 NEURAL TRAJECTORY ANALYSIS

The dimensionality reduction and smoothing method, Gaussian-Process Factor Analysis (GPFA) Yu et. al.(2009) was used for extracting low-dimensional neural trajectories from a population of LSTM activation of each recurrent layer. The GPFA model was fitted to time-series data of LSTM activation of a recurrent layer by the EM algorithm. We preprocessed time-series data of LSTM activation with the median filter of size of three in order to remove synchronous activation of many neurons that causes the rank deficiency of observation covariance matrix and prevents the use of GPFA.

## 3 RESULTS

We used five different priming conditions with four different priming sequences of different writers chosen from the validation dataset, one condition involving no priming, and three different target text strings: "The Whole Earth Catalog,", "stay hungry, stay foolish." and "It was their farewell message". These target texts did not appear in the training and the validation datasets. We collected 8 sets of network outputs and recorded the LSTM activation of the three recurrent layers from the RNN model for each condition with different seeds for the random number generators. For each layer, a GPFA model was fitted to recorded LSTM activation of 120 trials covering all priming and text conditions.

Figure 1 shows representative examples of GPFA neural trajectories of each recurrent layer during the synthesis. Generated handwriting samples show the difference in the writing styles between writers. The neural trajectories of each recurrent layer show larger difference among handwritings generated from different writing styles than those from the same style (Figure 2). These difference are especially obvious in the second layer, but statistically significant in the first and third layers, too (Mann-Whiteny U-test, $p<0.001$). These results suggest that different writing styles are encoded in different subspaces in the neural space of the RNN model and the second recurrent layer is important for maintaining a writing style during the synthesis of handwriting.

Next, we investigated the encoding of a character in the subspaces of the neural space. Figure 3 shows parts of neural trajectories during synthesis of a particular character. In the first and third layers, different characters give different neural trajectories in the subspace. These results suggest that different characters are represented as state dynamics in the subspace, especially in the first and third recurrent layers.

## 4 CONCLUSION

By analyzing the neural trajectories we investigated the internal representations of the three-layered recurrent neural network model for handwriting synthesis. Our visualizations of neural trajectories show that different writing styles are encoded in different subspaces inside the internal neural space

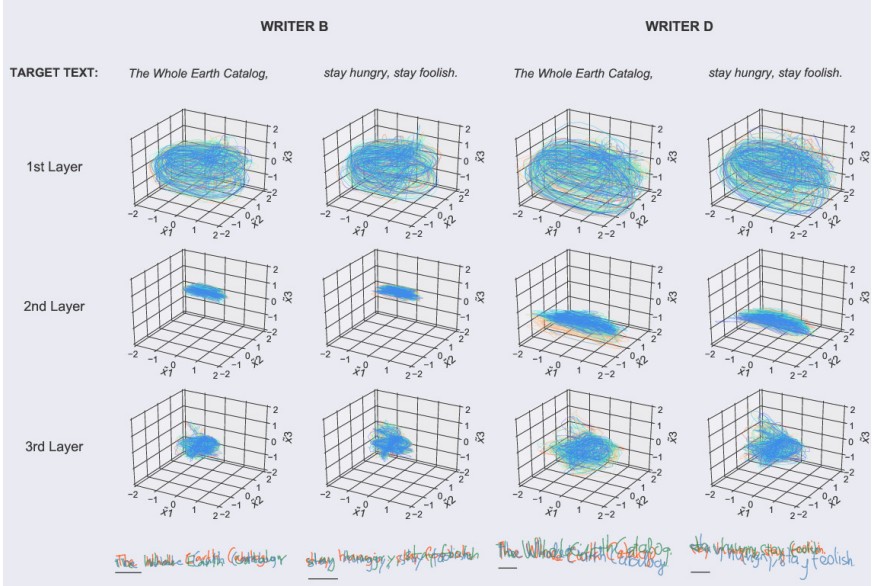

Figure 1: The three rows above show the top three dimensions of the orthonormalized GPFA neural trajectories for 8 different samplings for a given priming condition and a target text noted on the top. Trajectories of different trials are represented by different colors. The bottom row shows 3 randomly selected samples of generated handwriting. Black horizontal bars indicate length of 5 units.

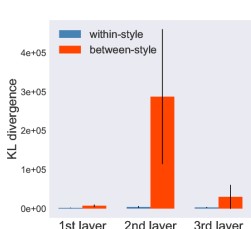

Figure 2: KL divergence between a pair of distributions of neural trajectories of different priming and text conditions.

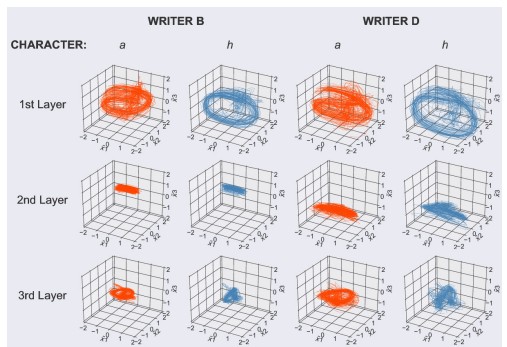

Figure 3: Each panel shows partial trajectories corresponding to character 'a' and 'h' collected from 24 samples of three different texts and a given style noted on the top.

where the information of strokes for a character is embedded. Moreover, our analysis suggests that each recurrent layer has a different role for the internal processes. These results demonstrate that the neural trajectory analysis is a very powerful method for intuitive understanding on how a recurrent neural network works.

ACKNOWLEDGMENTS

We thank Chris Garry for proofreading for and discussions about our manuscript.

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
