# OpenReview forum: "Neural Trajectory Analysis of Recurrent Neural Network In Handwriting Synthesis"
_ICLR.cc/2018/Workshop — Reject_

### Official Review · AnonReviewer2 · 2018-03-02
**Interesting analysis of low-dimensional trajectories in the internal state space of LSTM recurrent networks that learned to generate handwriting.**

**Rating:** 8
**Confidence:** 5

**Review:**

Interesting analysis of low-dimensional trajectories in the internal state space of LSTM recurrent networks that learned to generate handwriting.

Are the authors really using the LSTM of 1997, or the LSTM variant by Gers et al (2000) with forget gates (now sometimes called gated recurrent units)? It's the 2000 variant that most people are using now through Tensorflow etc.

Should be accepted after minor revisions.

---

### Official Review · AnonReviewer1 · 2018-03-09
**Sound approach but not sure about the visualization**

**Rating:** 3
**Confidence:** 4

**Review:**

The authors of this paper attempt to analyze the state dynamics of deep LSTM networks using the GPFA model. The method is applied to a deep LSTM network that was trained on handwriting synthesis task. As a workshop paper, I think their approach is sound and interesting, and the choice of the task is also nice since abundant amount of meta data is available in IAM-OnDB dataset.

I think the paper could have made more stronger points if it didn't only show how trajectories can look different when the priming conditions and target texts are different but also show what are shared within the same priming conditions and target texts. Also, analyzing which factor is more dominant and why. It is a bit obvious to me that the neural trajectories will look different when the conditions are different and also depending on which layer. Thus, I don't agree (yet) that neural trajectories have added more intuitive understanding on recurrent neural networks from this work.

---

### Decision · Program_Chairs · 2018-03-20
**ICLR 2018 Workshop Acceptance Decision**

**Decision:**

Reject

**Comment:**

Based on the reviews, this paper has not been accepted for presentation at the ICLR workshop. However, the conversation and updates can continue to appear here on OpenReview.